# First Phylogeny of Bitterbush Family, Picramniaceae (Picramniales)

**DOI:** 10.3390/plants9020284

**Published:** 2020-02-21

**Authors:** Alexey Shipunov, Shyla Carr, Spencer Furniss, Kyle Pay, José Rubens Pirani

**Affiliations:** 1Minot State University, Minot, ND 58707, USA; 2University of São Paulo, São Paulo 01000-000, Brazil

**Keywords:** *Alvaradoa*, *Nothotalisia*, *Picramnia*, Sapindales, rosids, *rbc*L, *trn*L-F, ITS

## Abstract

Picramniaceae is the only member of Picramniales which is sister to the clade (Sapindales (Huerteales (Malvales, Brassicales))) in the rosidsmalvids. Not much is known about most aspects of their ecology, geography, and morphology. The family is restricted to American tropics. Picramniaceae representatives are rich in secondary metabolites; some species are known to be important for pharmaceutical purposes. Traditionally, Picramniaceae was classified as a subfamily of Simaroubaceae, but from 1995 on, it has been segregated containing two genera, *Picramnia* and *Alvaradoa*, with the recent addition of a third genus, *Nothotalisia,* described in 2011. Only a few species of the family have been the subject of DNA-related research, and fewer than half of the species have been included in morphological phylogenetic analyses. It is clear that Picramniaceae remains a largely under-researched plant group. Here we present the first molecular phylogenetic tree of the group, based on both chloroplast and nuclear markers, widely adopted in the plant DNA barcoding. The main findings are: The family and its genera are monophyletic and *Picramnia* is sister to two other genera; some clades corroborate previous assumptions of relationships made on a morphological or geographical basis, while most parts of the molecular topology suggest high levels of homoplasy in the morphological evolution of *Picramnia*.

## 1. Introduction

Most flowering plants families described by the end of the 18th century have proven to be robust, stable groups. However, molecular tools have also been used to find less stable assemblages. One example of this is Simaroubaceae, a family in the Sapindales which was long suspected of being “non-natural” [1], and finally, in 1995 the two genera then forming subfamilies Picramnioideae and Alvaradoideae, *Picramnia* Sw. and *Alvaradoa* Liebm., were separated as Picramniaceae family [2].

Picramniaceae Fernando and Quinn is a rosid family ofca. 50 known species. They are dioecious, neotropical plants, with alternate, pinnately compound leaves, minute flowers with stamens opposite the petals, and a syncarpous gynoeceum with an inconspicuous style [3]. It is the only member of Picramniales which is robustly supported [4] as a sister group to the clade (Sapindales (Huerteales (Malvales, Brassicales))) [5]. Both family and order are restricted to American tropics, with centers of diversity in Mexico, the Amazonian region, and Southern Brazil (Figure 1). Whereas not much is known about many aspects of their life, Picramniaceae representatives are rich in secondary metabolites, and some species are known to be important for pharmaceutical purposes (as anti-malaria and anti-cancer drugs), even though their chemistry is not thoroughly studied [6].

Due to their small, actinomorphic flowers, bitter bark, and pinnately compound leaves, they were initially placed in Simaroubaceae [7]. However, the structure of flowers within Picramniaceae is seriously different [1,2]; the most striking is their syncarpous ovary with 2–3 ovules per locule while gynoecium of Simaroubaceae is typically almost apocarpous, with 1–2 ovules per carpel. 

Picramniaceae was formerly circumscribed with two genera, *Picramnia* [8] and *Alvaradoa* [9]. *Picramnia* is small or mid-size trees that grow either in the lower stories of moist tropical forests (e.g., in Amazonia and Choco) or in more dry habitats, from Mexico and Caribbeans to Southern Brazil and Northern Argentina. Some of them (like the northernmost *Picramnia pentandra* Sw. from South Florida) frequently grow as shrubs. Gross morphology of *Picramnia* is rather uniform and non-distinct, this was probably a reason of some notorious mistakes. One of examples is that Hispaniola generic endemic *Casabitoa perfae* described as a plant with putative affinities to Phyllanthaceae, is actually identical to *Picramnia dictyoneura* (Urb.) Urb. and Ekman [10].

*Alvaradoa* is significantly more xerophilous and shrubby, with wind-dispersed fruits (contrary to the animal-dispersed fruits of *Picramnia* and *Nothotalisia*). The range of distribution of the genus is conspicuously disjunct: Mexico and Caribbeans (Cuba, Hispaniola, and Jamaica), and Bolivia, Argentina, and South Brazil, as depicted by Thomas [11].

Recently, the third genus, *Nothotalisia* W.W.Thomas [12], was discovered among herbarium samples of *Talisia*Aubl. (Sapindaceae). *Nothotalisia*, like some species of *Picramnia*, includes mid-size forest trees that exhibits a broad distribution, from Panama to Peru, and likely associated with wet tropical forests of Choco and Amazonia. Interestingly, *Nothotalisia* appears not to be rare in collections: For example, it is abundant in herbarium samples collected in the Darien province of Panama (42 samples only in MO herbarium, dated from 1959 to 2011). In all, we believe that diversity of Picramniaceae is far from being exhausted, and we can always expect new discoveries.

We should mention here also the monotypic *Gumillea* Ruiz and Pav. [13] which might also belong to Picramniaceae [14]. This genus was described from some unknown place in Peru and unfortunately not recollected so far. In general, *Gumillea auriculata* Ruiz and Pav. reminds *Picramnia* but the plant bears stipules and bisexual flowers. Even though there is no recent specimen of that plant nor any molecular data, the morphology of inflorescence, flowers, and leaves suggests a relationship to Picramniales. The structures described and illustrated as “stipules” look like pseudostipules found in some species of *Picramnia* (e.g., *P. campestris* Rizzini and Occhioni*, P. guianensis* (Aubl.) Jans.-Jac.) where petiole is reduced and modified basal leaflets deflected to protect axillary buds [15,16].

Only a few species of Picramniaceae have been the subject of DNA-related research, and there are no molecular phylogenetic analyses so far. No generic subdivision was established; even the total number of species is not absolutely clear. Apparently, Picramniaceae remains an under-researched group.

To fill this gap, we performed a broad sampling of the group, aiming at collecting as many tissue samples as possible to amplify several widely used DNA barcoding markers in order to cover the species diversity in the group. We believe that our reconstruction of phylogenetic trees may become a basis for future studies in the group.

## 2. Material and Methods

We collected 276 tissue samples (Appendix A) from representatives of Picramniaceae and some other species (used as outgroups in the following analyses) and produce the first molecular phylogenetic trees of the group.

Our strategy was developed with an idea of the broadest sampling. We typically obtained multiple samples per species and attempted to extract DNA and sequence our markers multiple times until we reach satisfactory results. As we work with under-studied group, it was also especially important to provide photographic vouchers for each sample, and to be sure of the correct identifications, so either determination from known experts or our own determinations were preferred. At the end of this process, only a few species were missed from our samples, and most of them are either rare or local. As of today, our sequencing list contains 41 species, which is about 80% of the supposed group diversity. Most of them are newly sequenced material, only eight species represented with information in the public databases (GenBank and BOLD). In all, we were able to increase the amount of available information five-fold. Due to the apparent problems with identification, we always trusted our samples first. To improve the quality of databases, we constructed the working classification of Picraminiaceae (Appendix A), which now also contains the results of this phylogenetic study.

DNA extraction of the Picramniaceae herbarium samples was not always straightforward. Less than 20% of herbarium samples, fresh or old, yielded the DNA of the appropriate quality. We believe that the main reasons are the conditions of their collection and preparation. We were able also to extract DNA from living samples (collected in silica gel or just air-dried) taken from 8 species, and in all these cases the DNA extraction, amplification and sequencing were successful.

The DNA extraction was made using NUCLEOSPIN Plant II Kit (MACHEREY-NAGEL GmbH & Co. KG, Düren, Germany), which we believe is a good trade-off between efficiency and simplicity. We improved the protocol to increase the lysis time to 30 min and employ thermomixer on the slow rotation speed (350 rpm) instead of a water bath. We used Nanodrop 1000 Spectrophotometer (Thermo Scientific, Wilmington, DE, USA), to estimate concentration and purity (the 260/280 nm ratio of absorbance) of DNA.

We employed three of the most widely used barcoding DNA markers, chloroplast *trn*L-F [17] and *rbc*L [18], and nuclear ITS. These short barcoding DNA markers are thought to be the best to amplify from herbarium samples.Typically, our reaction mixture had a total volume of 20 μL which contained 5.2 μL of PCR Master Mix (components mostly from Thermo Fisher Scientific, Waltham, Massachusetts supplied with Platinum DNA Taq Polymerase; this was chosen after the series of experiments with other polymerases), 1 μL of 10 μM forward and reverse primers, 2 μL of DNA solution from the extraction and 10.8 μL of MQ purified water (obtained from a Barnstead GenPure Pro system, Thermo Scientific, Langenselbold, Germany) in the TBT-PAR water mix [19]. The latter was specifically developed to improve amplification from the “recalcitrant” herbarium samples. Thermal cycler programs were mostly 94° for 5 min, then 35 cycles of 94° for 1 min; 51° (or similar, depends on the primer) for 1 min, 72° for 2 min, and finally 72° for 10 min. PCR products were sent for purification and sequencing to Functional Biosciences, Inc. (Madison, WY, USA) and sequenced there following standard Sanger-based protocol. Sequences were obtained, assembled, and edited using Sequencher™ 4.5 (Genes Codes Corporation, Ann Arbor, MI, USA). This is how we ended with almost 780 sequencing chromatograms, of which we selected the 140 with the highest signal intensity for the next steps. These sequences represented 33 plant species.

To make this research portable and expandable, we automated most of the steps with the“Ripeline” workflow (Figure 2). This is the collection of UNIX shell and R [20] scripts that automate many processes related with sequence selection, quality checking, alignments, gap coding, concatenation, and phylogenetic tree production. Ripeline involves multiple pieces of software, for example, AliView [21], MUSCLE [22], APE [23], MrBayes [24], shipunov [25], and phangorn [26]. The Ripeline example, which uses its essential features and includes associated R code and documentation, is freely available from the Github [27].

With the help of Ripeline, we were able to create the super-matrix, which includes all three DNA markers and obtain maximal parsimony (MP) and Bayesian (MB) phylogeny trees. Maximum parsimony analyses were run with the help of R phangorn package [26] using parsimony ratchet [28] with 2000 iterations and then 1000 bootstrap replicates. Bayesian analyses were run through the combination of MrBayes 3.2.6 [24] and R shipunov [25] packages. MCMC chains were run for 1,000,000 generations, sampling every 10th generation resulting in 100,000 trees. The first 25% of trees were discarded as burn-in, and the remaining trees were summed to calculate the posterior probabilities. Trees were rooted with representatives of Sapindales (one out of the four orders forming the sister group of Picramniales) as outgroups: *Leitneria floridana* Chapm. (currently included in Simaroubaceae), *Kirkia acuminata* Oliv. (Kirkiaceae) and *Talisia nervosa* Radlk. (Sapindaceae), or with *L. floridana* alone. To stabilize the tree and to increase branch support, we used the hyper-matrix approach [29] and concatenated several alternative alignments of the ITS data part.

No morphological matrix is available for the whole family, so one of the most straightforward ways to assess its morphological diversity was to use the hierarchical components of identification keys [30]. To see how well our results correspond with the accumulated knowledge about the family, we employed these keys (along with data from morphological descriptions when available) from the three publications covering the diversity of Picramniaceae in Central America [31], Western South America [32], and Brazil [15]. We also used geographical data to see if any patterns of that type might be observed on our phylogeny trees. Ancestral character estimation was done with the help of APE package [23]; we encoded polymorphisms as most ancestral states.

Datasets, scripts, trees for individual markers and other information used in preparation of this publication are available as one zip archive from the first author’s Open Repository [33]. We encourage readers to reproduce our results and develop our methods further. All sequences were deposited into the GenBank (Appendix A).

## 3. Results

### 3.1. Picramniaceae in General

With the outgroups mentioned above, Picramniaceae formed a stable clade. First branches (Figure 3) are *Picramnia* and *Nothotalisia* + *Alvaradoa*, with sufficient (close to 100%) support.

#### 3.1.1. Alvaradoa

This part of the tree is strongly supported (Figure 4). South American *Alvaradoa subovata* Cronquist and *A. puberulenta* (Monach.) Sleumer form the first clade whereas West Indian species—the second clade. *Alvaradoa amorphoides* Liebm., which grows in Mexico and Florida, is basal to West Indian species.

#### 3.1.2. Nothotalisia

*Nothotalisia peruviana* (Standl.) W.W. Thomas and *N. cancellata* W.W. Thomas form the stable clade to (Figure 4) which *N. piranii* W.W. Thomas is the sister group.

#### 3.1.3. Picramnia

Generally, there is much less stability in that part of the tree. Many clades are not reliably supported. The most supported are the following groupings: (1) *Picramnia antidesma* Sw. + *P. deflexa* W.W. Thomas+ *P. latifolia* Tul. + *P. thomasii* Gonzalez-Martinez and J. Jimenez Ram.; (2) *P. andrade-limae* Pirani + *P. elliptica* Pirani and W.W. Thomas + *P. bahiensis* Turcz. + *P. grandifolia* Engl. (latter two with less confidence); (3) *P. dictyoneura* Planch. + *P. sphaerocarpa* Planch. and (4) *P. glazioviana* Engl. + *P. parvifolia* Engl.; (5) *P. oreadica* Pirani + *P. ramiflora* Planch. + *P. campestris* + *P. juniniana* J.F.Macbr. However, some lower supported groups also make sense, especially (6) *P. excelsa* Kuhlm. ex Pirani + *P. gracilis* Tul. + *P. nuriensis* Steyerm. +*P. tumbesina* Cornejo.

### 3.2. Morphological and Geographical Patterns

Even after extensive sampling, some species of *Picramnia* are still missing in our molecular dataset. To test the possible placement of these species, we used the hierarchical component of identification keys together with phylogeny trees and employed *k*-nearest neighbor machine learning with bootstrap to point on possible sister species. Since super-species level classification is not developed, we decided to return trios of putative neighbor species which the highest bootstrap confidence (Table 1).

There are multiple morphological and geographical patterns observed on the ancestral character estimations (Figure 5a–f). For example, 3–4 carpels recognized as the most ancestral type (Figure 5a), whereas monocarpellate (*Alvaradoa*) and bicarpellate (part of *Picramnia*) are likely advanced character states. There is no visible pattern for the position of inflorescence (Figure 5b). Reversely, the phylogenetic distribution of the inflorescence type is rich in patterns (Figure 5c), but this ancestral type is not well defined. Ancestral merosity was estimated as 5–6 merous, whereas tetramerous or trimerous flowers evolved independently in several *Picramnia* clades (Figure 5d). A type of trichome (Figure 5e) significantly deviates from the ancestral “attenuate” type only twice within the genus. Finally, regional geography estimation (Figure 5f) suggests domination of the Central American and West Indies regions, with more recent origin of two other areas (West South American and Amazonian, and Bolivian plus extra-Amazonian Brazil) centers. Remarkably, parts of the topology seem to be more or less geographically structured: two clades are mostly composed by Bolivian to extra-Amazonian Brazilian species; one clade is mostly composed by species ranging from Darien (Panama) to Peru extending to the Amazonian region; and two other clades composed of a mixture of species either showing the latter distribution pattern or species ranging from Central American to the West Indies (Figure 5f).

## 4. Discussion

### 4.1. Picramniaceae in General

Picramniaceae is here supported with three monophyletic genera. The monophyly of each genus was expected since they are clearly distinct taxa on morphological grounds. However, the molecular evidence of a closer relationship between *Nothotalisia* and *Alvaradoa* seems unnatural as we compare the general morphology of the threegenera. As Thomas [12] already pointed out, “*Nothotalisia* is strikingly similar to many species of *Picramnia*” since both genera bear alternate leaflets with oblique bases and attenuate tips, racemose inflorescences with congested lateral cymes, and berry-like fruits. *Alvaradoa* is easily distinguished from the former two genera by its much smaller, oblong leaflets with symmetric bases, and flattened, winged, dry fruits. On the other hand, *Nothotalisia* is unique in having an androgynophore on male flowers; this peculiar structure bears the five stamens united by their filaments and a small pistillode on top.

### 4.2. Alvaradoa and Nothotalisia

Within *Alvaradoa*, the Caribbean clade is well supported, whereas *A. amorphoides* + South American species are less robust. Nevertheless, this latter clade might point to the concurrent hypotheses on the causes of the disjunct pattern between Central America and subtropical South American Cone, observed in several groups of angiosperms (e.g., [34]): ancient connection (vicariance) vs. long-distance dispersal. A reappraisal of such an American Amphitropical disjunct pattern was recently carried out by Simpson et al. [35], suggesting dispersal as the leading cause in most of the lineages. Within *Nothotalisia*, Amazonian *N. piranii* is distinct from the more widespread *N. cancellata* and *N. peruviana*.

### 4.3. Picramnia

Several of the smaller clades within *Picramnia* seem to be strongly correlated with morphology and congruent with geography. Other clades are composed by a mixture of species that had not so far been considered as closely related on a morphological basis.

Pirani [15,16] proposed that *P. ramiflora* and *P. latifolia* could be related since they share several morphological features, and some herbarium specimens may be hard to ascribe to one or the other species. However, they emerge far away in the molecular topology. *P. latifolia* presents as main diagnostic characters distinguishing it from the remaining species the mostly lateral inflorescences bearing (sub)sessile flowers and fruits, although pedicellate flowers occur in some specimens. It is a widespread, mostly Amazonian species that reach Central America; therefore, its closer relationship to Central American to Andean species (*P. antidesma, P. deflexa,* and *P. thomasii)* makes sense on a geographical basis. *P. ramiflora,* however, is a quite disjunct species from the Brazilian Atlantic Forest.

The clade formed by (*P. thomasii (P.deflexa (P.latifolia,* and *P.antidesma)))* is morphologically heterogeneous, with a mixture of species bearing pentamerous flowers (*P. latifolia* and *P. thomasii*) and species with trimerous or tetramerous flowers (the other two). This fact implies that even though the numbers of sepals and petals have a consistent relevance for species recognition within the genus, often used in keys (e.g., [15,31,36]), it seems to hold little phylogenetic signal. Other examples of clades containing either trimerous, tetrameous or pentamerous species are: (a) *P. guerrerensis* W.W. Thomas (5-merous) nested with *P. hirsuta* W.W. Thomas and *P. teapensis* Tul. (3–4-merous); (b) *P. elliptica* and *P. grandifolia* (5-merous) closely related to *P. bahiensis* and *P. andrade-limae* (3-merous); (c) *P. excelsa* (3-merous) nested within a clade with *P. sellowii* and *P. guianensis* (both 5-merous), along with *P. sphaerocarpa* which is particularly polymorphic producing 3–4–5-merous flowers.

Other morphological features that have long been used in the taxonomy of *Picramnia,* as well as to generate assumptions of relationships, concern to the inflorescence type and position (e.g., [15,31]); these characters may be critically considered, as indicated by the molecular data. For example, *P. hirsuta,* with long simple racemes, maybe sister to *P. polyantha* (Benth.) Planch.*,* which bears profusely branched thyrses. In the same way, racemose *P. elliptica* and *P. grandifolia* here emerged within a clade with *P. andrade-limae* and *P. bahiensis;* both provided with thyrses.

Nonetheless, some assumptions of relationships previously made on morphological grounds are here corroborated by the molecular data. For example, the close relationship between *P. glazioviana* and *P. parvifolia* proposed by Pirani [15], based on several features, was here recovered; and both are partially sympatric species of the Atlantic Forest in Eastern to Southern Brazil. However, it seems awkward the fact that a species so similar to *P. parvifolia* as is *P. excelsa* [15], both sympatric inside the *Araucaria* forests of Southern Brazil, did not emerge in that same clade. Such reasoning may also be pointed to *P. glazioviana,* which is quite similar and sympatric to *P. ciliata* in the Brazilian coastal moist forests*,* but this latter emerged near the base of the molecular tree. *P. andrade-limae* and *P. bahiensis* also have trimerous flowers with peculiar obcordate petals and assumed as related to the aforementioned Atlantic Forest group with *P. glazioviana* [15], but emerged on other branches of the phylogenetic tree.

Astonishingly, the two samples of *P. oreadica* did not emerge together, but the fact that each of them is currently ascribed to distinct allopatric subspecies may be considered here [15]. These are heliophilous shrubs inhabiting the open *cerrados* of Central Brazilian Plateau [16], known so far from few herbarium specimens, and our results show that further studies on taxonomic circumscription are here necessary. Also, it should be expected that *P. guianensis* would be nested in the clade including *P. oreadica* and *P. campestris*, because they share a special type of indumentum on the ovary (peculiar clavate trichomes), and are all mostly related to sandy or rocky substrates. These species live mostly as heliophytic shrubs, contrasting with the typical understory habit prevailing in the genus. However, the fact the *P. guianensis* emerged in another clade, that feature may have appeared independently. *P. ferrea* Pirani and W.W.Thomas*,* unsuccessfully sampled in this analysis, could probably belong to the *P. guianensis* clade on a biogeographical basis, or the *P. campestris* clade on morphological grounds (e.g., clavate trichomes on ovary surface).

Hence, there is some evidence which corroborates previous assumptions of relationships made on a morphological basis, while most parts of the molecular topology suggest high levels of homoplasy in the structural evolution of *Picramnia*. On the other hand, as we pointed out previously, it is remarkable that parts of the topology seem to be more or less geographically structured since two clades are mostly composed of Bolivian to extra-Amazonian Brazilian species. Another clade is mostly composed of species ranging from Darien (Panama) to Peru extending to the Amazonian region. In contrast, the next two clades are composed of a mixture of species, either showing the latter distribution pattern or species ranging from Central American to the West Indies. Even though we are aware that the definition of these areas of distribution is somewhat artificial, they are mostly in agreement with biogeographical regions and provinces previously proposed for the Americas [27]. The spatial component revealed in our phylogenetic results suggests that the diversification of some of the lineages within *Picramnia* may reflect some of the geomorphological, climatic, and ecological grounds that helped to shape or drive the history of the neotropical biota.

#### Final Remarks

Even with extensive sampling and continuous sequencing efforts, our data is still incomplete. Only 20% of our sequencing efforts ended with reliable sequences. GenBank data generally lack sequences of Picramniaceae. Some rare species are still not sampled.

Nevertheless, we think that our phylogeny results might serve as a backbone for future research. Our best hope is that this work represents the beginning of the second, molecular stage in Picramniaceae research. One of the most apparent next steps is to check how morphology and DNA go together, the molecular “weight” of traditional morphological characters.

## Figures and Tables

**Figure 1 plants-09-00284-f001:**
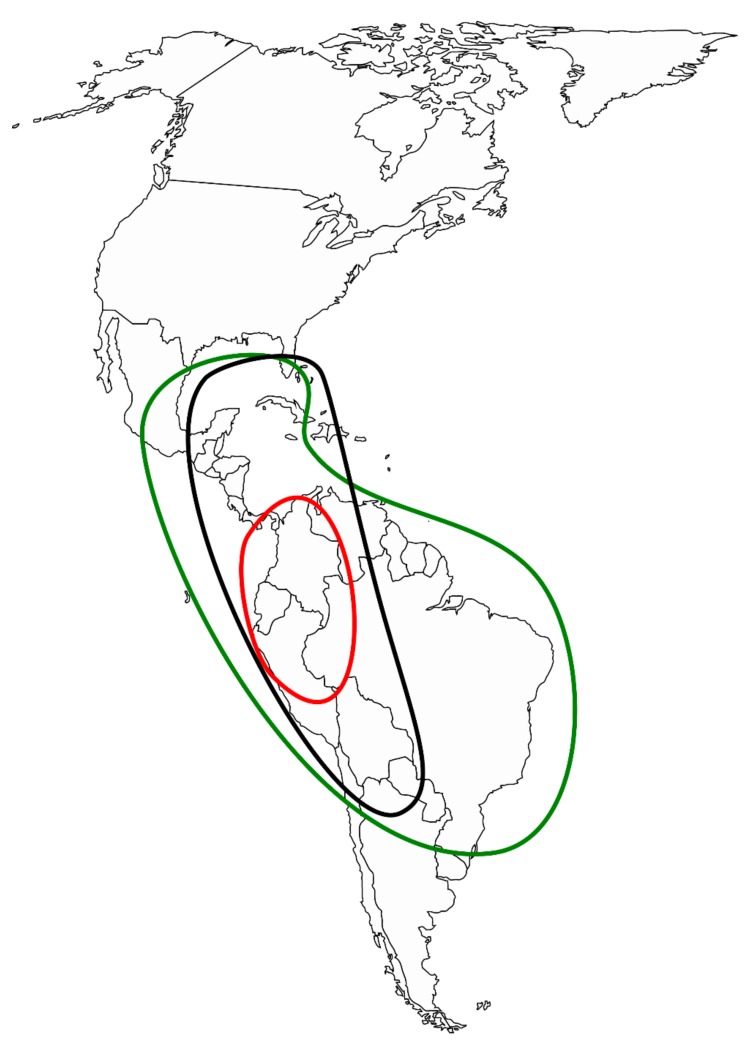
Geographic distribution of *Picramnia* (green), *Alvaradoa* (black), and *Nothotalisia* (red).

**Figure 2 plants-09-00284-f002:**
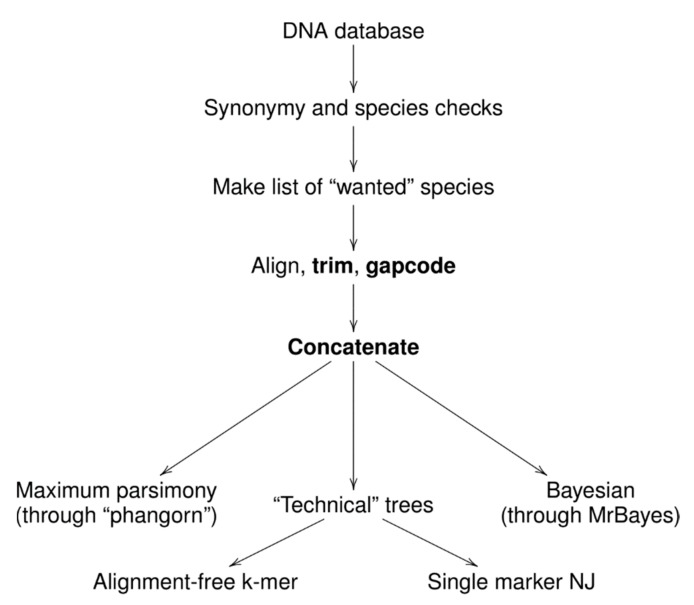
Ripeline: workflow and basic features.

**Figure 3 plants-09-00284-f003:**
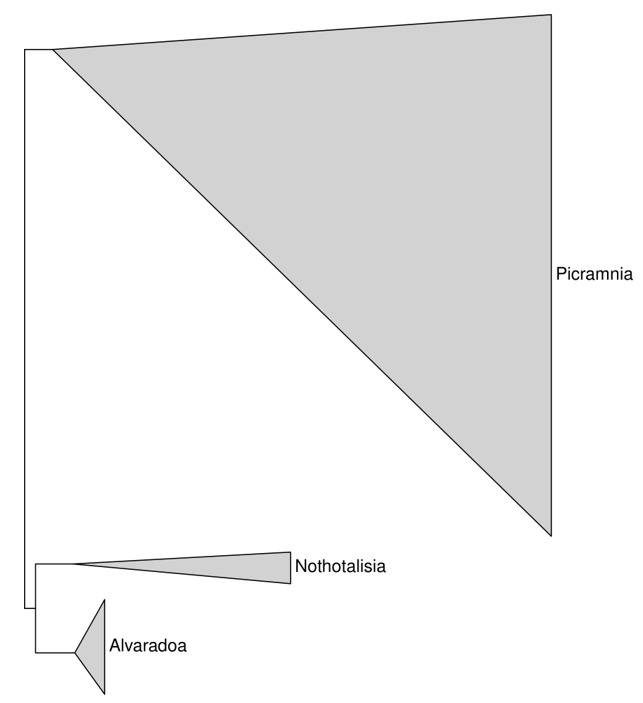
The overview of our phylogeny trees depicting the relationships and relative size of the three strongly supported monophyletic genera of Picramniaceae. Each triangle is the result of concatenation applied to the branches of the phylogenetic tree.

**Figure 4 plants-09-00284-f004:**
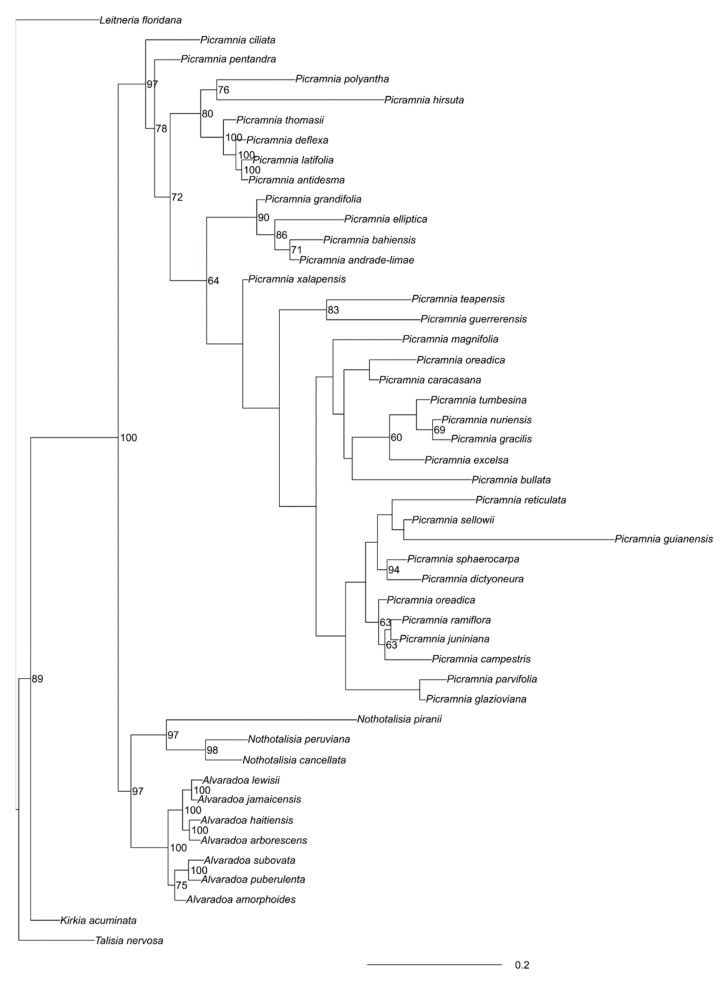
MrBayes tree of concatenated (“semi-strict”) loci.Numbers are Bayesian support (when ≥ 60%).

**Figure 5 plants-09-00284-f005:**
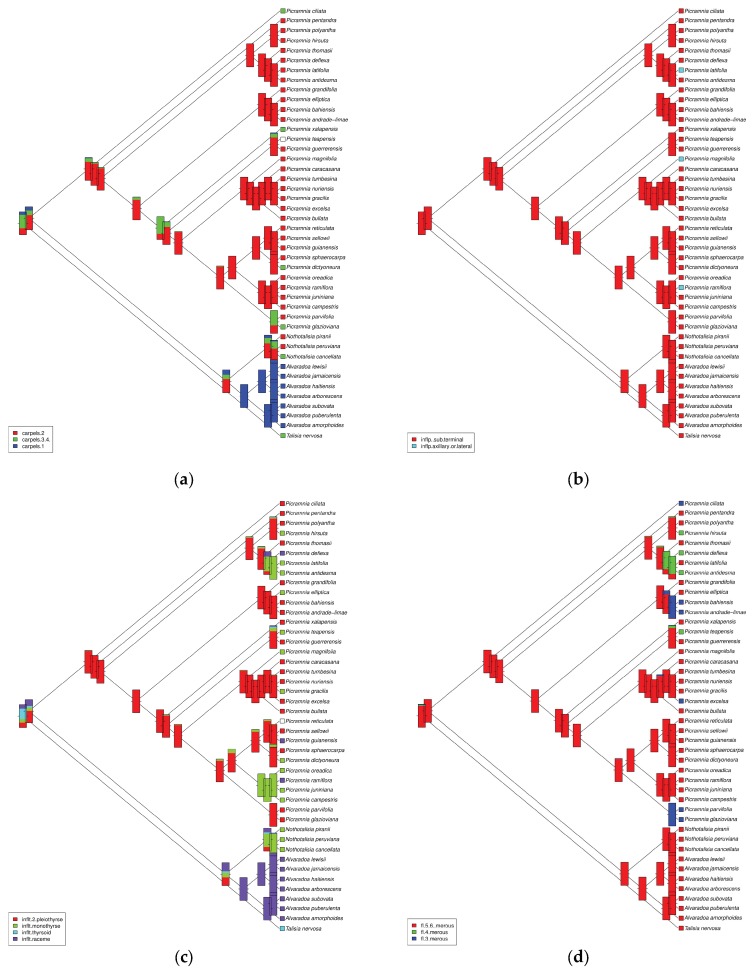
Ancestral character estimation on six different types of characters: (**a**) number of carpels, (**b**) position of the inflorescence, (**c**) type of the inflorescence, (**d**) merosity of flower, (**e**) type of trichomes, (**f**) geography on the regional level.

**Table 1 plants-09-00284-t001:** Placements of *Picramnia* species based on *k*-nearest neighbor machine learning.

Species to Place	Most Likely Neigbor	Probability, %	Most Likely Neigbor	Probability, %	Most Likely Neigbor	Probability, %
*P. coccinea*	*P. latifolia*	31	*P. caracasana*	18	*P. juniniana*	16
*P. ferrea*	*P. juniniana*	18	*P. oreadica*	15	*P. campestris*	14
*P. gardneri*	*P. latifolia*	32	*P. caracasana*	19	*P. juniniana*	14
*P. gardneri*	*P. caracasana*	34	*P. latifolia*	27	*P. sellowii*	27
*P. grandiflora*	*P. latifolia*	26	*P. juniniana*	15	*P. oreadica*	15
*P. matudai*	*P. gracilis*	17	*P. hirsuta*	15	*P. teapensis*	14
*P. spruceana*	*P. latifolia*	36	*P. caracasana*	27	*P. sellowii*	20
*P. spruceana*	*P. caracasana*	18	*P. sellowii*	17	*P. latifolia*	15

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
