# Peer review of "First Phylogeny of Bitterbush Family, Picramniaceae (Picramniales)"

_plants, 2020, doi:10.3390/plants9020284_

Round 1

Reviewer 1 Report

This is a nice article representing a wealth of information about a poorly studied plant group, the family Picramniaceae. Despite being a separately standing group that comprises its own order, Picramniales, and a subject of a recent discovery of a new genus, Nothotalisia, Picramniaceae were outside of molecular studies. So the article is very timely and important. 

I have only few minor suggestions. First, you mention the problems with DNA extraction and sequencing (especially with sequencing - only 20% of the efforts yielded satisfactory results). What is the reason (or at least the manifestation - because probably you do not know the reason) of these problems? The absence of amplification? (might be helpful to try another polymerases - I have no doubt that Platinum Taq is not the best one). The mixed signal on chromatograms? It would be very helpful, in particular for the  researchers that may want to continue and to develop the molecular studies on Picramniaceae, to know what exactly are the difficulties that you faced with, and what optimizations you tried in order to overcome them (if any). It would also be helpful to characterize the divergence of markers that you used, within and between the genera of Picramniaceae, and to provide the trees for individual markers, not only for concatenated dataset. What is the reason of low support values within Picramnia? Was there a conflict between topologies inferred from nuclear and plastid loci? Or just the lack of information?  Second, you need to provide the accession number for the sequences generated in this study. I expected to see them in the supplementary table 1 but they are not there. 

Reviewer 2 Report

General comments and remarks.

The authors attempt to give a first overview of the phylogenetic relationships based on molecular data, of Picramniaceae, a family of tropical species of the Americas, relatively recently separated from Sapindaceae.

The paper has clear goals, it is well written and easily understood by the reader. In the phylogenetic tree, however, it is attempted, perhaps a little hurriedly, and possibly without sufficient molecular data, to form groups within the genus Picramnia. These groups do not seem to be sufficiently supported or at least this is not clear in the tree where in many places the confidence levels of the branches are absent. The reason is not mentioned in the text. Is the support level too low (less than 50%)? If so, there should be a reference to the phylogenetic tree's caption as well as in the text.

I believe that the phylogenetic tree should not lead to conclusions about possible clusters and affinities within the genus Picramnia. At least not at this stage. A larger number of samples may be necessary to arrive at safer and more satisfactorily supported conclusions. My suggestion is to distinguish groups in a less detailed way or not to distinguish groups in this paper. This distinction could be made in a later approach, and if the authors have more data in their hands. This need is also emphasized by the authors themselves, in several parts of their text (many clades are not reliably supported) and particularly at the end of the paper.

In the other two genera of the family, it seems that things are easier and clearer and that the conclusions that the authors draw are fully supported by molecular data.

Another point that is not clear in the paper is how easy it is to distinguish the species of Picramnia if someone considers only morphological characters. In some parts of the text, there are references for species that are not easy to distinguish.

Comments and remarks on specific parts of the text 

Line35: Both family and order are restricted …. 

Figure 1: It should be stated which line corresponds to the distribution of which genus. Perhaps different colours or types of lines (eg dotted instead of consecutive ones) help the reader to better understand the limits of the distribution of the 3 genera.

Lines 51, 53, 54 and elsewhere in the text: Please cite the name of the author for all the species, when these are first mentioned in the text.

Line 65: Why the genus Grumillea is mentioned. Is there any suspicion that it might belong to the family Picramniaceae?

Line 111: Typically, our PCR … maybe a word is missing here

Lines 122-123: …we selected the best 140 for the next steps... Could you please be more specific? What criteria did you use for this distinction?

Lines 140-141: Why did you choose these species as outgroups? What was the rationale?

Line 156: Please define sufficient support. There is no reference in the text to determine the appropriate level of support.

Figure 4: I do not think that in the Picramnia the proposed groups are well supported by the results. In some cases, the level of support is very low (60%), while in others it is not mentioned at all. The whole genus Picramnia could be given without any distinction of groups at this stage. It is clear after all that even the morphological characters used by the keys to distinguish species do not seem to correspond to the proposed groups of the tree. Perhaps a more general revision of the genus is necessary and could be the subject of a separate paper. 

Other questions

P. polyantha and P. hirsuta could be included in the first group? 

In the second group, what does Picramnia elliptica Minas means? Is Minas a region? And why do you separated it from P. elliptica in the same group?

What is Picramnia sp. Nadruz? A species that you have collected and have not classified into a known species?

Table 1: Species names should be written in a single row. Since they all belong to the genus Picramnia only the initial letter can be given

Figure 5: These charts are impossible to read. You should make then bigger and use larger characters.

Support table 2: I would suggest not just by mentioning the names of the family species. You can improve the table by adding other data such as their total distribution, their habitat preferences, their altitudinal range etc. 

Reviewer 3 Report

The manuscript by Shipunov et al. presents the first comprehensive phylogeny of Picramniaceae. It an interesting study with a strong underlying approach to generating an initial phylogeny. Although this is an interesting group that merits further investigation, the manuscript in its current form needs some minor revision particularly improvements in figures and writing. I will comment on the figures below but the writing revisions are too numerous to list and I suggest some serious writing revision before publication. All my comments are as follows:

Abstract:

Lines 11-12: very awkward sentence, would suggest rewording. Also just say species instead of representatives.

Line 20: just write that Picramnia is sister to the other two genera

Introductions:

General: How old is this group?

Line 48 “are small or”

Lines 52-54: can be more concise.

Material and Methods:

Line 82: “We collected 276 ..”

Line 85: don’t follow this: I assumed your sampling strategy but be based on the group you are studying and not your extraction strategy ….

Line 93, what databases and accessions? For what species and why use these if you only trust your own identifications?

Line 103: Would just state the kit and modifications and leave out interpretations of efficiency and simplicity

Line 105-107: just state that a nanodrop was used to estimate concentration and purity especially since that nanodrop has a fairly significant margin of error

Line 109-110 - needs a reference.

Line 122-123 – odd statements, seems to be more like a burden? Why not just state we selected 140 sequences of sufficient quality for analysis; also mention the number of species represented by this number of sequences.

Lines – why not post on github or dryad?

Line 152: accessions?

Lines 170 – 175: I feel like this is a consequence of poor taxonomic keys, which is not surprising but should be considered here and in the discussion.

Lines 177 -182: Although very neat, it seems to suggest additional markers are needed! This should again be included in the discussion in more detail. This is a great first step but you should discuss future prospects.

Line 249: need rephrasing.

Line 291-293 – needs rephrasing.

Figure 1: needs a legend or some explanation of the colours. (i.e. link them to species).

Figure 2: The workflow is ok and a little too simple looking. Details can be added with respect to alignment, trimming etc.

Figure3 - not very informative, can be cleverly incorporated with figure 4

Figure 4- some numbers and species names intersect with lines. Why not add geography? And the moderate support (89) for the family suggests a need for better outgroup sampling.

Figure 5 – Way too small, the legend is illegible and again these can be simplified and made to look far better. Adding geography if it makes sense would be good as well.
